# Structure and function of the divalent anion/Na$^+$ symporter from *Vibrio cholerae* and a humanized variant

Rongxin Nie[1], Steven Stark[1], Jindrich Symersky[1], Ronald S. Kaplan[1] & Min Lu[1]

Integral membrane proteins of the divalent anion/Na$^+$ symporter (DASS) family translocate dicarboxylate, tricarboxylate or sulphate across cell membranes, typically by utilizing the preexisting Na$^+$ gradient. The molecular determinants for substrate recognition by DASS remain obscure, largely owing to the absence of any substrate-bound DASS structure. Here we present 2.8-Å resolution X-ray structures of VcINDY, a DASS from *Vibrio cholerae* that catalyses the co-transport of Na$^+$ and succinate. These structures portray the Na$^+$-bound VcINDY in complexes with succinate and citrate, elucidating the binding sites for substrate and two Na$^+$ ions. Furthermore, we report the structures of a humanized variant of VcINDY in complexes with succinate and citrate, which predict how a human citrate-transporting DASS may interact with its bound substrate. Our findings provide insights into metabolite transport by DASS, establishing a molecular basis for future studies on the regulation of this transport process.

[1] Department of Biochemistry and Molecular Biology, Rosalind Franklin University of Medicine and Science, 3333 Green Bay Road, North Chicago, Illinois 60064, USA. Correspondence and requests for materials should be addressed to M.L. (email: min.lu@rosalindfranklin.edu).

**M**embrane transporters belonging to the ubiquitous divalent anion/Na$^+$ symporter (DASS) family move Krebs cycle intermediates or sulphate across the cell membrane typically by utilizing the preexisting Na$^+$ gradient[1–3]. In particular, mammalian DASS proteins NaCT, NaDC1 and NaDC3 can mediate the symport of three or more Na$^+$ ions and C$_6$-tricarboxylate (such as citrate) or C$_4$-dicarboxylate (such as succinate), whereas NaS1 and NaS2 transport sulphate[4–8]. The bacterial DASS proteins, by contrast, catalyse the coupled uptake of two or more Na$^+$ ions and C$_4$-dicarboxylate, rather than C$_6$-tricarboxylate or sulphate[9–13].

Since citrate, succinate and other C$_4$-dicarboxylates are important intermediates and/or regulators of the energy metabolism, the modulation of DASS function expectedly impacts fatty acid synthesis, energy expenditure and life span. Particularly, a handful of mutations found in a DASS-encoding gene nearly doubled the average adult life span in fruit flies, likely by promoting a metabolic state that mimics caloric and dietary restriction[14,15]. In addition, the knockdown of either one of the genes encoding NaDC2 and NaCT in worms decreased their body size and fat content, and/or increased their life span[16,17]. More recently, deletion of the gene-encoding NaCT protected mice from the adiposity and insulin resistance induced by high-fat feeding and aging[18]. Altogether, mounting evidence supported the utility of DASS proteins, particularly NaCT, as therapeutic targets for tackling obesity, type 2 diabetes and other metabolic diseases[1,2].

Despite such clinical significance, the molecular mechanism underlying DASS-mediated co-transport remains poorly understood. The 3.2-Å resolution crystal structure of citrate-bound VcINDY, a DASS from *Vibrio cholerae*, elucidates the transporter architecture and implicates the amino acids in Na$^+$ and citrate binding[19]. However, like other well-characterized bacterial DASS proteins, VcINDY is known to transport succinate and other C$_4$-dicarboxylates, rather than citrate (C$_6$-tricarboxylate)[13,19].

Furthermore, although VcINDY was suggested to catalyse the co-transport of three or more Na$^+$ ions and C$_4$-dicarboxylate[13], only one Na$^+$-binding site was observed crystallographically in the published work[19]. Although a second Na$^+$-binding site was predicted[19], no direct structural evidence was available and the assignment of this predicted site remained speculative. Therefore, previous studies fell short of addressing a central question in DASS-mediated transport: how does the transporter recognize substrate and multiple Na$^+$ ions?

Herein we present the structure of succinate-bound VcINDY at a resolution of 2.8 Å, which, to our knowledge, offers the first molecular view of a substrate-bound DASS. Our data enabled us to elucidate a previously undiscovered Na$^+$-binding site in VcINDY as well as how this transporter utilizes helix dipoles and Na$^+$ binding to interact with trans-C$_4$-dicarboxylate. Furthermore, we report the structure of a citrate-bound VcINDY alongside those of the succinate- and citrate-bound MT5, a humanized variant of VcINDY, all determined to 2.8-Å resolution. Collectively, our results bring to light how citrate inhibits VcINDY-mediated succinate transport and how a DASS distinguishes between C$_4$-dicarboxylate and C$_6$-tricarboxylate, thereby recasting the conceptual framework for understanding how a DASS specifically recognizes and transports its anionic substrate.

## Results

**Structure of a substrate-bound VcINDY.** We crystallized VcINDY in the presence of succinate and Na$^+$, and determined the structure by combining molecular replacement and multiple isomorphous replacement and anomalous scattering (MIRAS) phasing (Table 1, Supplementary Table 1). Although not essential for solving the phase problem, the MIRAS phases allowed us to model substantially more amino acids than what had been achieved previously[19] (445 versus 398 out of 462 residues), to

## Table 1 | Data collection and refinement statistics.

|  | Succinate-bound VcINDY | Citrate-bound VcINDY | Citrate-bound MT5 | Succinate-bound MT5 |
|---|---|---|---|---|
| *Data collection* |  |  |  |  |
| Space group | $P2_1$ | $P2_1$ | $P2_1$ | $P2_1$ |
| Cell dimensions |  |  |  |  |
| $a,b,c$ (Å) | 106.68, 101.91, 167.74 | 106.13, 102.11, 167.99 | 106.09, 101.54, 168.89 | 107.14, 102.28, 170.86 |
| $\alpha,\beta,\gamma$ (°) | 90, 98.97, 90 | 90, 99.52, 90 | 90, 99.73, 90 | 90, 98.31, 90 |
| Resolution (Å) | 100-2.80 (2.85-2.80) | 100-2.80 (2.85-2.80) | 100-2.80 (2.85-2.80) | 100-2.80 (2.85-2.80) |
| $R_{sym}$ | 0.10 (0.51) | 0.10 (0.49) | 0.11 (0.54) | 0.11 (0.60) |
| $I/\sigma$ | 21.0 (2.0) | 19.7 (1.8) | 27.6 (2.1) | 26.2 (1.9) |
| Completeness (%) | 99.9 (99.8) | 92.5 (58.1) | 99.5 (92.5) | 96.4 (61.8) |
| Redundancy | 12.9 | 15.2 | 27.1 | 24.5 |
|  |  |  |  |  |
| *Refinement* |  |  |  |  |
| Resolution range (Å) | 15.0-2.80 | 15.0-2.80 | 15-2.80 | 15-2.80 |
| No. reflections | 75,407 | 73,983 | 75,382 | 77,152 |
| $R_{cryst}/R_{free}$ (%) | 26.1/27.0 | 24.2/26.1 | 25.2/26.8 | 24.7/26.6 |
| No. atoms | 13,388 | 13,408 | 13,456 | 13,436 |
| $\langle B \rangle_{protein}$ | 89 | 76 | 78 | 102 |
| $\langle B \rangle_{ligand}$ | 88 | 126 | 76 | 99 |
| $\langle B \rangle_{ion}$ | 80 | 82 | 72 | 94 |
| r.m.s.d. |  |  |  |  |
| Bond lengths (Å) | 0.006 | 0.006 | 0.006 | 0.005 |
| Bond angles (°) | 1.0 | 1.1 | 1.1 | 1.0 |
| Ramachandran |  |  |  |  |
| Favoured (%) | 99.0 | 99.0 | 99.2 | 99.4 |
| Allowed (%) | 1.0 | 1.0 | 0.8 | 0.6 |
| Disallowed (%) | 0 | 0 | 0 | 0 |

r.m.s.d., root mean squared deviation.

locate the bound succinate with confidence, and to compare different co-structures without interference from model bias. Moreover, the density-modified MIRAS maps revealed that parts of the published VcINDY structure[19], which are directly related to the binding of Na$^+$ and citrate, had been incorrectly determined (see below). Using the updated structures and resulting $F_o$–$F_c$ electron density maps the other co-structures were well-determined by and also supported by the unbiased MIRAS maps.

Our structure reveals a homodimeric arrangement (Fig. 1a), with each VcINDY protomer comprising eleven membrane-spanning helices (TM1–TM11), two re-entrant helix-turn-helix hairpins (HP$_{in}$ and HP$_{out}$) and two interfacial helices (H4c and H9c; Supplementary Fig. 1). Viewed in parallel to the membrane, the VcINDY dimer is shaped like an inverted bowl with its concave mouth facing the cytoplasm, allowing the aqueous solution to reach the midpoint of the membrane (Fig. 1b). In each VcINDY protomer, the inter-helical loops in HP$_{in}$ and HP$_{out}$, and the intra-helical loops within the discontinuous TM5 and TM10 meet approximately halfway across the membrane, forming a cleft that opens toward the cytoplasm.

**Na$^+$-binding sites in VcINDY.** Within this cleft, the binding sites of two Na$^+$ ions, designated as Na1 and Na2, were observed (Supplementary Fig. 2). We assigned the Na$^+$-binding sites based on the following lines of evidence. First, the electron density ascribed to Na$^+$ is surrounded by electronegative oxygen atoms, which is consistent with a cation coordination shell. Second, Na$^+$ was used throughout the VcINDY purification and was the only inorganic cation included in the crystallization solution. Third, the distances between the ions and liganding atoms range from 2.3 to 2.7 Å, which are appropriate for Na$^+$ coordination but too short for H-bonds formed between water molecules and VcINDY[20]. Indeed, we performed the valence test on the two sites, which is suggestive of the binding of Na$^+$ ions rather than water molecules[21]. Fourth, the two putative cations are penta-coordinated, which represents one of the most common Na$^+$ coordination arrangements[20]. CheckMyMetal, a structure validation server for metal-binding sites[22], supported the assignment of Na1 and Na2 as the binding sites for Na$^+$, rather than K$^+$ or Ca$^{2+}$. Fifth, a single amino-acid change in either Na1 or Na2 affected the Na$^+$-dependence of VcINDY-mediated transport (see below).

Previously, Na$^+$ in Na2 was not identified in VcINDY (ref. 19), likely due to errors in model building (Supplementary Figs 3 and 4). In each of the two Na$^+$-binding sites now

elucidated in VcINDY, Na$^+$ is penta-coordinated to two amino-acid side-chain and three backbone carbonyl oxygen atoms. Specifically, Na$^+$ in Na1 is coordinated to the side-chain hydroxyl of S146 and side-chain carbonyl of N151, in addition to the main-chain carbonyls of S146, S150 and G199 (Fig. 2a); whereas Na$^+$ in Na2 binds to the side-chain hydroxyl of T373 and side-chain carbonyl of N378, besides the backbone carbonyls of T373, A376 and A420 (Fig. 2b). Overall, the two Na$^+$ ions are separated by > 13 Å and bound to the pseudo-symmetry-related HP$_{in}$ and HP$_{out}$, respectively. However, this symmetry is broken when detailed coordination interactions are inspected. In particular, S150 in HP$_{in}$ binds to Na$^+$, whereas its counterpart in HP$_{out}$, that of S377, does not. In addition, I149 in HP$_{in}$ makes no contact with Na$^+$, but its equivalent in HP$_{out}$, A376, does.

**Functional importance of the Na$^+$-binding sites.** Moreover, the mutations of S146, N151 and N378, impaired the transporter activity of VcINDY in a whole-cell based assay, even though these mutations did not substantially affect the expression level of the transporter[19]. Furthermore, we replaced S146 and T373 individually with alanine and reconstituted the detergent-purified mutants into proteoliposomes. We then compared the function of the mutants with that of the wild type transporter. Under optimal conditions, VcINDY exhibited robust succinate transport activity in a counter-flow assay[13] (Fig. 3a). Our kinetic studies on VcINDY revealed that the $K_M$ and $V_{max}$ for succinate transport are 0.22 mM and 254 nmol mg$^{-1}$ min$^{-1}$, respectively. At succinate concentration below the $K_M$, we found the VcINDY-catalyzed transport to be highly dependent on Na$^+$ concentrations, with a Hill coefficient of 2.7 and a $K_{0.5-Na}$ of 15 mM, respectively (Fig. 3b). Significantly, mutations of S146 and T373 increased the $K_M$ for succinate transport by more than 11- and 1,227-fold and elevated the $K_{0.5-Na}$ to 37 and > 800 mM, respectively (Fig. 4).

Our data suggested that VcINDY binds at least two Na$^+$ ions during transport and that Na1 and Na2 play pivotal roles in the succinate transport. Indeed, mutation S146A appeared to diminish the ability of membrane-embedded VcINDY to bind Na$^+$, because higher concentrations of Na$^+$ were required by the mutant to effect maximal stimulation of succinate transport than VcINDY (Fig. 4b). In addition, mutation T373A abolished the ability of Na$^+$ to stimulate VcINDY-mediated transport within the tested Na$^+$ concentration range (Fig. 4b). Furthermore, mutation S146A or T373A raised the $K_M$ for succinate, albeit to a different extent (Fig. 4a), likely by indirectly weakening the binding of substrate to VcINDY within the membrane, as

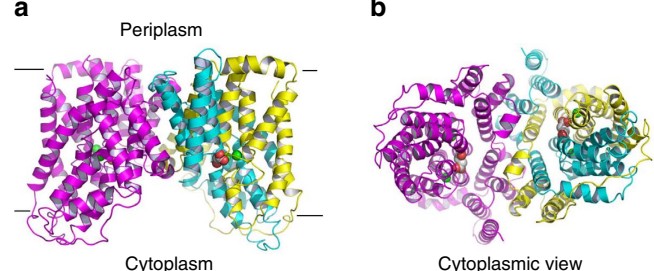

**Figure 1 | Structure of the succinate-bound VcINDY.** (**a**) Structure of dimeric VcINDY as viewed from the membrane bilayer. VcINDY is shown in ribbon rendition, the N (18–231) and C (232–462) domains in one protomer are coloured cyan and yellow, respectively, whereas the other protomer is coloured magenta. Na$^+$ ions (green) and succinate are drawn as spheres. (**b**) The cytoplasmic view of the VcINDY structure, highlighting the solvent-accessible succinate and buried Na$^+$ ions.

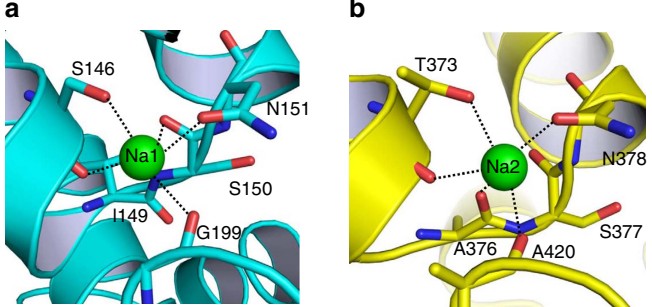

**Figure 2 | Close-up views of the Na$^+$-binding sites in VcINDY.** (**a**) Structure of the Na$^+$-binding site in the N domain. (**b**) The previously unobserved Na$^+$-binding site within the C domain. Na$^+$ ions are drawn as green spheres and relevant amino acids as stick models. Dashed lines indicate the coordination interactions.

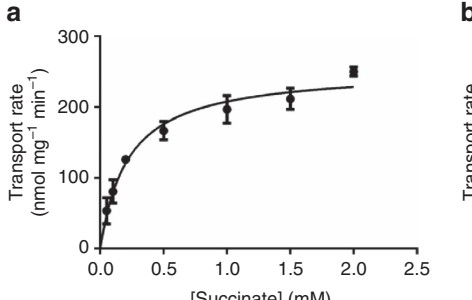
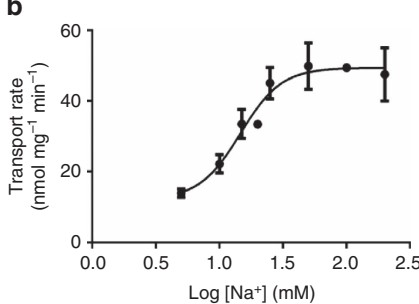

**Figure 3 | Succinate transport activity of VcINDY.** (**a**) The initial rates of succinate transport at 200 mM external Na$^+$ and pH 7.4 were plotted against external succinate concentrations. Data were averaged to fit to the Michaelis–Menten equation, yielding a $K_M$ of 0.22 mM and a $V_{max}$ of 254 nmol mg$^{-1}$ min$^{-1}$, respectively. (**b**) The initial rates of succinate transport at 50 µM external substrate concentration ($\sim$20% of the measured $K_M$) and pH 7.4 were plotted against the common logarithm (Log) of external Na$^+$ concentrations. Data were averaged to fit to the Hill equation, yielding a Hill coefficient of 2.7 and a $K_{0.5\text{-}Na}$ of 14.6 mM, respectively. The error bars represent s.d. from at least three independent experiments.

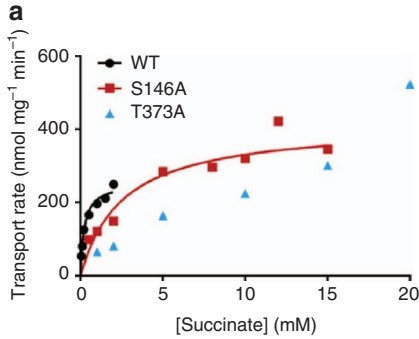
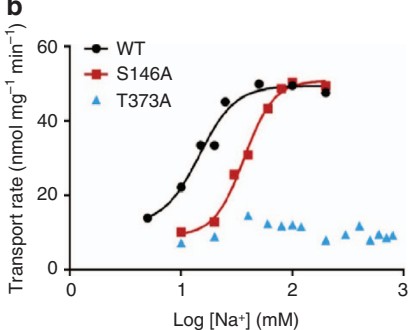

**Figure 4 | Mutational effects on the transport properties of VcINDY.** (**a**) The initial rates of succinate transport at 200 mM external Na$^+$ and pH 7.4 were plotted against external succinate concentrations. The data for S146A were averaged to fit to the Michaelis–Menten equation, yielding a $K_M$ of 2.57 mM and a $V_{max}$ of 416 nmol mg$^{-1}$ min$^{-1}$. Transport for T373A remained unsaturable up to 20 mM succinate, suggesting a $K_M$ of >270 mM and precluding an accurate measurement of $K_M$ or $V_{max}$. (**b**) The initial rates of succinate transport at 200 µM external substrate concentration ($\sim$8% of the $K_M$ for S146A) and pH 7.4 were plotted against the common logarithm of external Na$^+$ concentrations. The data for S146A were averaged to fit to the Hill equation, yielding a Hill coefficient of 3.2 and a $K_{0.5\text{-}Na}$ of 36.9 mM. T373A-mediated transport of succinate lacked Na$^+$-dependence within the tested range (up to 800 mM), suggesting that the $K_{0.5\text{-}Na}$ for T373A is >800 mM. Similar observations were made with 500 µM ($\sim$20% of the $K_M$) and 4 mM external succinate for S146A and T373A, respectively. At 500 µM succinate, the data for S146A revealed a Hill coefficient of 3.1 and a $K_{0.5\text{-}Na}$ of 47.6 mM. For comparison, the data for VcINDY were plotted to highlight the functional consequences of mutations in Na1 and Na2.

Na$^+$ coordination stabilizes the bound succinate in VcINDY (see below).

Of particular interest, T373A had more pronounced effect on the transport function than the structurally analogous S146A (Fig. 2), arguing that Na2 has a more important role than Na1 during transport and the two sites are functionally non-equivalent. In addition, mutations of human NaDC3$^{S143}$, NaDC3$^{N144}$ and NaDC3$^{N484}$, which are equivalent of VcINDY$^{S150}$, VcINDY$^{N151}$ and VcINDY$^{N378}$, respectively, exerted deleterious effects on the transport function[23]. In particular, NaDC3$^{S143A}$ exhibited impaired transport function, whereas NaDC3$^{N144A}$ or NaDC3$^{N484A}$ had almost no measurable transporter activity. These data further suggested that Na1 and Na2 are preserved in NaDC3 and our structure is a valid model for studying the mechanism of human DASS.

**Succinate-binding site in VcINDY.** Within the Na$^+$-binding cleft in VcINDY, the electron density for succinate was also observed (Supplementary Fig. 5). The bound succinate makes contacts with VcINDY mainly through H-bonding interactions and is partly exposed to cytoplasm (Fig. 1b), indicating that the transporter adopts an inward-open conformation. Specifically, the side-chain amide of N151 and the side-chain hydroxyl of T152 donate H-bonds to one carboxylate in succinate, whereas the

side-chain hydroxyl S377 and the side-chain amide of N378 make contacts with the other succinate carboxylate (Fig. 5a). Moreover, the side chain of T379 interacts with the aliphatic portion of the succinate through van der Waals interactions, and S200 and T421 form H-bonds with N151 and N378, respectively, which may further stabilize the interactions between succinate and VcINDY.

Significantly, the bound succinate adopts an extended conformation, structurally mimicking fumarate, a C$_4$-dicarboxylate with a central *trans* double bond. This structural mimicry implied that fumarate interacts with VcINDY similarly to succinate, thus explaining why fumarate effectively inhibits VcINDY-mediated succinate transport[13,19]. By contrast, maleate, the *cis* isomer of fumarate, exerts less inhibitory effect on the succinate transport than fumarate[13], arguing that VcINDY is specific for C$_4$-dicarboxylate in a stretched conformation. Notably, most of the succinate-interacting amino acids are conserved, implying that the preference for trans-dicarboxylate is not limited to VcINDY but also shared by other DASS proteins.

Furthermore, the alanine substitution of N151 or N378 reduced the binding of succinate to VcINDY and gave rise to severely impaired transporter activity[19]. By contrast, the mutation of T379, which makes van der Waals interactions with the succinate, had only moderately deleterious impact on the transport function[19]. However, T379 was suggested to play a role in substrate recognition[19] and is often replaced by valine in

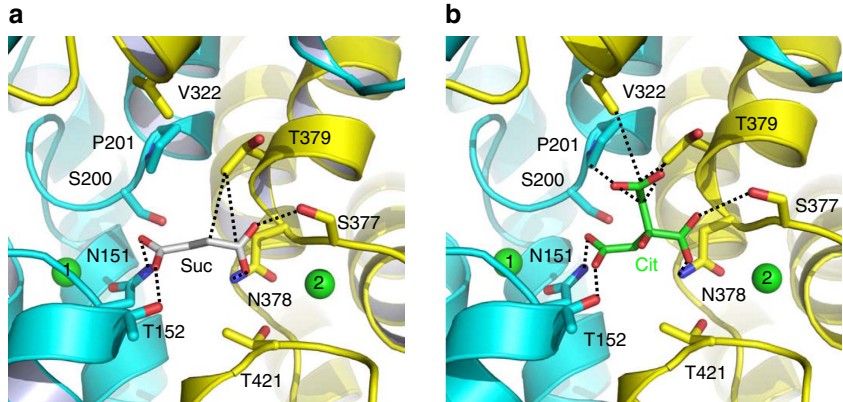

**Figure 5 | Close-up views of the succinate- and citrate-binding sites in VcINDY.** (**a**) Structure of the succinate-binding site. (**b**) Detailed view of the citrate-binding site. Succinate (grey), citrate (green) and relevant amino acids are drawn as stick models, whereas the Na$^+$ ions are shown as green spheres. Dashed lines highlight the interactions between VcINDY and succinate or citrate.

human DASS. Moreover, the individual mutations of NaDC3$^{N144}$, NaDC3$^{T145}$, NaDC3$^{T253}$, NaDC3$^{S483}$, NaDC3$^{N484}$ and NaDC3$^{T527}$ (equivalent of VcINDY$^{N151}$, VcINDY$^{T152}$, VcINDY$^{S200}$, VcINDY$^{S377}$, VcINDY$^{N378}$ and VcINDY$^{T421}$) all affected the transport function[23]. In particular, NaDC3$^{T253S}$ and NaDC3$^{S483A}$ exhibited suppressed transport function, whereas the transporter activity of NaDC3$^{N144A}$, NaDC3$^{T145}$, NaDC3$^{S483A}$, NaDC3$^{T527N}$ was almost completely abolished. Altogether, these data suggested that the observed succinate-binding site in VcINDY is functionally relevant and conserved within the DASS family.

**Insights into the symport mechanism.** Previous studies suggested that Na$^+$ ions bind to DASS before it can bind its substrate[9–12], implying that the coordination of Na$^+$ promotes substrate binding. In the succinate-binding VcINDY, the Na$^+$ ions in Na1 and Na2 coordinate several succinate-binding amino acids and thus stabilize the conformation of these amino-acid side chains. This arrangement helps to explain why the transport of succinate and Na$^+$ is strictly coupled, as they bind to a common subset of amino acids, and the binding or unbinding of one likely affects that of the other. Moreover, the Na$^+$ ions may attract succinate, which carries two negative charges[13], through long-range electrostatic interactions within the low-dielectric intramembrane environment[24,25]. Furthermore, the amino ends of four short helices from HP$_{in}$, HP$_{out}$, TM5 and TM10, which possess localized positive dipoles, all point toward the bound succinate and stabilize its two negative charges (crystallization pH ∼ 7). Notably, the stabilization of negative charges by the opposing, positive helix dipoles within inverted structural repeats may represent a general theme for achieving anion selectivity within the lipid bilayer by membrane proteins[26–29].

The succinate-bound VcINDY structure also implies that the substrate is released from the intracellular-facing transporter prior to the dissociation of Na$^+$ ions from Na1 and Na2, since the bound substrate is more accessible to the cytoplasm than the two Na$^+$ ions (Fig. 1b). Furthermore, Na2 appears functionally more important than Na1, implying that the binding of Na$^+$ to Na2 in an outward-facing VcINDY precedes as well as promotes the Na$^+$ coordination at Na1. Previous studies also implied that VcINDY catalyses the co-transport of three or more Na$^+$ ions and succinate[13], although only two Na$^+$-binding sites are apparent in our structure. One possibility is that one or more Na$^+$ ions may have already been released from VcINDY before it

adopts the current inward-open conformation and evaded detection by our structural study.

**Structure of the citrate-bound VcINDY.** To uncover how VcINDY distinguishes between C$_4$-dicarboxylate and C$_6$-tricarboxylate, we determined the structure of a citrate-bound VcINDY (Table 1, Supplementary Table 2). Citrate inhibits the VcINDY-mediated succinate transport, likely as a competitive inhibitor[13,19]. We found the citrate-bound VcINDY structure to be similar to the succinate-bound form, with a root mean squared deviation (r.m.s.d. of < 1 Å for 445 Cα positions. In the citrate-bound VcINDY, the side-chain amide of N151 and the side-chain hydroxyl of T152 donate H-bonds to a terminal carboxylate (pro-R) of the bound citrate[30], whereas the side-chain hydroxyl of S377 and the side-chain amide of N378 form H-bonds with the central carboxylate of citrate (Fig. 5b). Moreover, the side chains of P201, V322 and T379 make van der Waals interactions with the bound citrate. One terminal carboxylate (pro-S) and the hydroxyl of citrate, however, make no interaction with VcINDY and project towards the solvent.

Notably, the positioning of citrate in VcINDY (Supplementary Fig. 6) is different from that in a published structure[19]. To examine whether this difference is real, we refined our structural model against the published X-ray data, which gave rise to better protein stereochemistry and a simultaneous drop in $R_{free}$ (> 2%). Significantly, our analysis suggested that the citrate had been incorrectly modelled previously (Supplementary Fig. 7). Despite the difference in citrate placement and incorrectly modelled amino acids (Supplementary Fig. 4), the two citrate-bound structures could be superimposed onto each other to yield a r.m.s.d. of < 1 Å for 393 common Cα positions, indicating that both structures captured VcINDY in the inward-open conformation.

More importantly, our co-structure indicated that the citrate- and succinate-binding sites overlap substantially in VcINDY, and that citrate inhibits the transport of succinate by preoccupying the substrate-binding site, that is, as a competitive inhibitor. In our VcINDY structures, the two carboxylates of succinate occupy virtually the same position as the central and pro-R carboxylates of citrate. This observation indicated that HP$_{in}$, HP$_{out}$ and the unwound region in TM10 constitute a 'trans-dicarboxylate-recognition' module in DASS. A prominent feature of this module is the absence of any protonatable or positively charged amino acids, starkly contrasting the succinate-binding water-

soluble proteins, in which positively charged Arg and Lys interact with the bound dicarboxylate[31–35].

*In vivo*, at least two carboxylates in citrate are deprotonated and negatively charged[13]. In the citrate-bound structure, the pro-S carboxylate, which probably carries a full negative charge (crystallization pH ~ 7), makes no interaction with VcINDY. Rather, P201, V322 and T379 pack against the bound citrate and steer the pro-S carboxylate away from the transporter. By contrast, both carboxylates in succinate are stabilized by the H-bonding interactions made with VcINDY. Therefore, the negative charges in citrate may not be fully 'neutralized' by its interactions with VcINDY in the membrane bilayer. This finding may explain why citrate is less effective in inhibiting VcINDY-mediated succinate transport than C4-dicarboxylates and why citrate preferably binds to the inward-facing VcINDY (refs 13,19). Assimilating the data from published studies[36,37] and our VcINDY structures, we argued that the interactions between citrate and the inward- or outward-facing VcINDY are similar but not identical: in particular, the pro-S carboxylate is less solvent-exposed in the outward-facing state and less stabilized by solvation than that in the inward-facing protein. Consequently, it would be energetically unfavourable to form the citrate-VcINDY complex in the membrane bilayer due to its negative charge surplus, especially in the outward-facing state.

**Structures of a humanized variant of VcINDY.** To validate this idea as well as to gain new insights into the mechanism of human DASS, we replaced eight amino acids surrounding the citrate-binding cleft by their counterparts in NaCT (Supplementary Fig. 1), which primarily transports citrate[6]. Of note, such a 'multi-mutational' approach was previously employed to study the interactions between antidepressants and a bacterial homologue of biogenic amine transporters[38]. We crystallized this octuple mutant of VcINDY, denoted MT5, in complex with citrate and determined the crystal structure (Table 1, Supplementary Table 3). Although the MT5 structure remains similar to that of VcINDY, one important difference centres on the pro-S carboxylate of the bound citrate (Supplementary Fig. 8). Specifically, the side-chain hydroxyl of S200T and the backbone amide of P201G in MT5 each donate an H-bond to the pro-S carboxylate, whereas the side chain of T379V makes van der Waals interactions with the citrate (Fig. 6a). In contrast to that in VcINDY, the pro-S carboxylate latches onto the amino ends of TM5b and the second helix in HP_out in MT5, with its putative negative charge stabilized by the positive helix dipoles (crystallization pH ~ 7). Since NaCT

transports the trianionic citrate[6], our structure may foretell the interactions between NaCT and its bound substrate.

We also determined the succinate-bound MT5 structure (Table 1, Supplementary Table 4), which revealed that MT5 binds succinate in the same way as VcINDY except that T379V in MT5 makes no contact with the bound substrate (Fig. 6b). Our transport assay further showed that MT5 retained the succinate-transporting activity, with the $K_M$ and $V_{max}$ values both similar to those of VcINDY (Supplementary Fig. 9). These data implied that T379 is not essential for the succinate transport and NaCT may also interact with a trans-dicarboxylate. Although VcINDY and MT5 interacted with succinate similarly, pronounced differences between them were found in the extent to which citrate inhibited the transporter-mediated uptake of succinate (Fig. 7). Specifically, 95 mM citrate reduced the succinate transport rate by < 20% in VcINDY, whereas a mere 10 mM citrate decreased the transport rate by > 40% in MT5 (at pH 7.4).

Since VcINDY and MT5 were inserted into the liposome membrane in presumably two orientations[13], our data suggested that the membrane-embedded MT5 interacts with trianionic citrate more strongly than VcINDY in both the inward- and outward-facing states. This stark difference is likely attributed to the altered pose of the pro-S carboxylate seen in MT5. In support of this notion, the temperature factors for the bound citrate in MT5 are lower than those in VcINDY (Table 1), arguing that the citrate was more firmly bound to MT5 than VcINDY at pH ~ 7. On one hand, such results should be viewed with the caveat that we were unable to elicit any appreciable citrate-transporting activity in VcINDY or MT5, which implied that further experiments would be required to convert MT5 into an efficient citrate transporter. On the other hand, given the amino-acid sequence similarity between VcINDY and NaCT as well as the homologue swap mutations carried by MT5, MT5 likely recapitulates the substrate-binding properties of NaCT considerably.

**Substrate-recognition modules in DASS.** Taken together, we assert that the amino ends of TM5b and the second helix in HP_out form yet another substrate-recognition module in DASS for differentiating C6-tricarboxylate from C4-dicarboxylate. In a C4-dicarboxylate-specific VcINDY, this module includes a Pro and a Thr (Fig. 8a), which selects against citrate by pushing away its pro-S carboxylate and likely giving rise to negative charge surplus within the membrane. In a C6-tricarboxylate-transporting NaCT, however, the Pro and Thr are superseded by Gly and Val, respectively, which enables the recognition of pro-S carboxylate via H-bonds and the stabilization of its potential negative charge

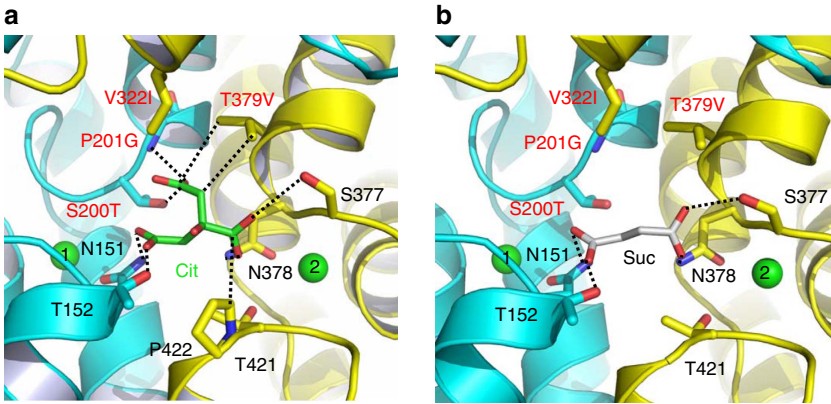

**Figure 6 | Detailed views of the citrate- and succinate-binding sites in MT5.** (**a**) Structure of the citrate-binding site. (**b**) Close-up view of the succinate-binding site. Citrate (green), succinate (grey) and relevant amino acids are drawn as stick models, while the Na+ ions are shown as green spheres. Humanizing amino-acid substitutions are highlighted in red. Dashed lines indicate the interactions between MT5 and citrate or succinate.

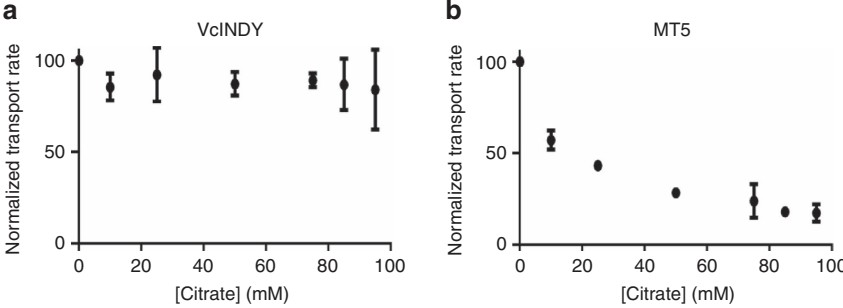

**Figure 7 | The inhibition of transporter-mediated transport by citrate.** Initial rates of succinate transport at $50 \, \mu M$ external substrate concentration, $200 \, mM \, Na^+$ and pH 7.4 were calculated as percentages of values for VcINDY (**a**) or MT5 (**b**) in the absence of added $K_3$-citrate. The normalized transport rates were plotted against the external citrate concentrations. Data were averaged and the error bars indicate s.d. from at least three independent experiments.

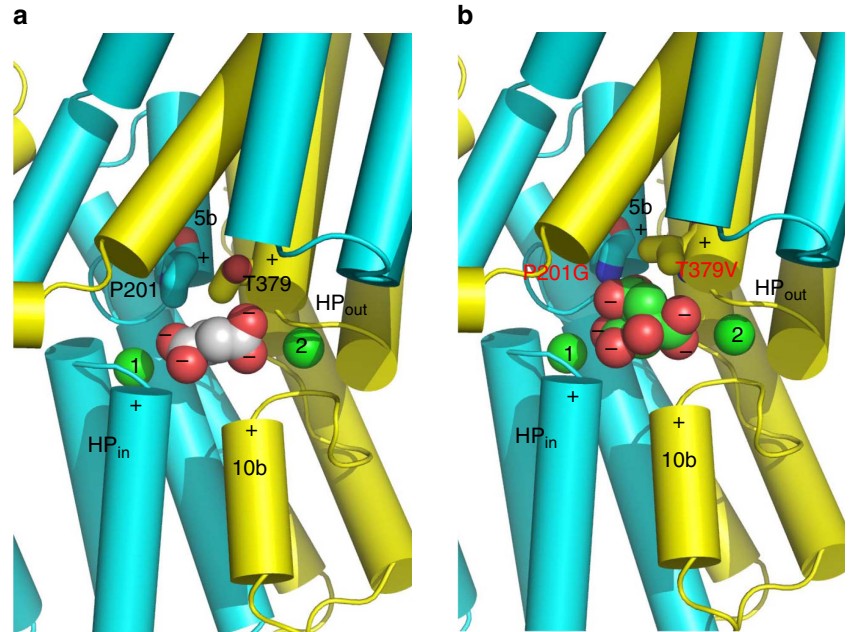

**Figure 8 | Structural basis for substrate recognition by DASS.** The N and C domains in VcINDY (**a**) and MT5 (**b**) are coloured cyan and yellow, respectively. Relevant amino acids are drawn as stick models, whereas the bound succinate (**a**) or citrate (**b**) as well as the $Na^+$ ions (green) are shown as spheres. Positive helix dipoles are highlighted by plus signs, whereas the negatively charged carboxylates in succinate or citrate are indicated by minus signs. The charged state of succinate or citrate is deduced based on the crystallization pH ($\sim 7$). Both the helix dipoles and $Na^+$ appear to contribute to the anion binding. Furthermore, P201 and T379 may enable VcINDY to select for succinate but against citrate. In MT5, P201 and T379 are replaced by Gly and Val, respectively, which may allow the membrane-embedded transporter to bind citrate more strongly than VcINDY.

(Fig. 8b). Thus, DASS appears equipped with two newfound substrate-recognition modules: one selective for trans-$C_4$-dicarboxylate and the other for $C_6$-tricarboxylate. Notably, $Na^+$ also contributes to the binding of $C_4$-dicarboxylate to DASS by stabilizing the first module. Our findings hence elucidate how a DASS recognizes its substrate and offer a new angle to understand protein-mediated anion transport in general.

## Discussion

As noted previously[19], the N and C domains of VcINDY are structurally related, with an r.m.s.d. of 3.5 Å for 101 common Cα positions, even though the two domains possess opposite membrane topology and share rather modest amino-acid sequence similarity (22% identity). Perhaps surprisingly, the transmembrane domain of VcINDY also exhibits striking structural resemblance to the dimeric AbgT transporters[39,40]

despite a lack of patent amino-acid sequence homology. Specifically, the structure of VcINDY can be superimposed onto those of AbgT to yield r.m.s.d. of 3.1 and 3.5 Å for 294 and 305 Cα atoms, respectively. Moreover, VcINDY bears 18 and 13% amino-acid sequence identity to the two AbgT transporters of known structure, respectively.

Interestingly, the two AbgT transporters appear to function as antibiotic efflux pumps[39,40] and are likely antiporters or exchangers, whereas most of the known DASS proteins are symporters[1,2]. The structural similarity between the AbgT and DASS protein families adds a new superfamily of secondary membrane transporters with shared dimeric organization and structural fold, notwithstanding their unrelated physiological functions and distinct transport mechanisms. Importantly, a $Na^+$-binding site similar to Na2 in VcINDY was also found in an AbgT transporter, and alanine substitutions of the cation-coordinating amino acids impaired the transport

function[40]. These data argued that at least one functionally important Na$^+$-binding site is conserved between the DASS and AbgT protein families.

Furthermore, the substrate-binding site in VcINDY lacks any positively charged amino acid, for example, Lys or Arg. DASS has evolved such a scheme probably because positively charged amino acids would discourage the binding of Na$^+$ in their vicinity due to electrostatic repulsion and/or cause the transporter to bind C$_4$-dicarboxylate much too tightly, thereby impeding the dissociation of substrate from the inward-facing transporter. Besides VcINDY, at least five membrane transporters of known structure also select for substrates that carry net negative charges: SeCitS from the citrate-sodium symporter family[37], Glt$_{Ph}$ and Glt$_{Tk}$ from the excitatory amino-acid transporter family[41,42], NarK and NarU from the nitrate/nitrite porter family[43–45]. In contrast to VcINDY, all these transporters utilize one or two Arg residues to bind the negatively charged groups in the substrate.

This seemingly unexpected difference may be understood in light of the substrate-binding site and/or the coupling mechanism. In Glt$_{Ph}$ and Glt$_{Tk}$, two Na$^+$-coupled symporters, an Asp residue is located in close proximity to the substrate-binding Arg residue. Therefore, the side-chain carboxylate of this Asp may neutralize the positive charge on the Arg side-chain and weaken the electrostatic attraction between the Arg and substrate, thereby facilitating the release of the negatively charged substrate. In NarK and NarU, both of which are exchangers, the negatively charged nitrate and nitrite can compete against each other for the two substrate-binding Arg residues, thus enabling the binding and unbinding of the substrate as well as the counter-transported ligand. Notably, these four proteins have entirely different fold from that of VcINDY, and their substrate-binding sites appear to have co-evolved with their transport mechanisms.

SeCitS, on the other hand, has a similar but not identical protein fold to that of VcINDY and catalyses the symport of Na$^+$ and citrate. Based on the published results, the transport mechanism of SeCitS may be similar to that of VcINDY[36,37]. However, the SeCitS-mediated transport of citrate is electroneutral[37], whereas VcINDY-catalyzed succinate transport appears electrogenic[13]. This difference implies that VcINDY interacts with three or more Na$^+$ ions while SeCitS binds only two Na$^+$ ions during transport. Probably, the higher number of co-transported Na$^+$ ions enables VcINDY to recognize and transport negatively charged substrates without engaging any positively charged Arg or Lys.

The lack of charged amino acid in the substrate- and Na$^+$-binding sites in VcINDY also begged the question of how a DASS stabilizes its unloaded or apo state. In Glt$_{Tk}$ and Glt$_{Ph}$, a substrate-binding Arg occupies the substrate-binding site in the empty transporter[46,47] and apparently acts as a substrate surrogate to stabilize the apo state. Upon the release of substrate and Na$^+$ ions into the cytoplasm, we suspect that water molecules may interact with the substrate- and/or Na$^+$-binding amino acids in VcINDY to stabilize its apo state, since both the substrate- and Na$^+$-binding sites are solvent-accessible in the inward-facing transporter (Fig. 1). In VcINDY, water molecules may be sufficient to compensate for the local charge imbalance caused by the dissociation of substrate and Na$^+$ ions, since no charged amino acid in the vacant substrate- and Na$^+$-binding sites needs to be 'neutralized'. If this notion holds true, then VcINDY may also behave as a water 'carrier' in the cell[48].

## Methods

**Protein expression and purification.** The genes encoding VcINDY and MT5 were synthesized (GenScript, NJ) and cloned into a modified pET28b vector with an N-terminal cleavable deca-histidine tag. Mutations were introduced into the gene-encoding VcINDY by the QuikChange method (Agilent Technologies) and were confirmed by DNA sequencing. Primers used include

5′-GCGTGACCGCTCTGCTGGCAATGTGGATCTCGAAC-3′ and 5′- GTTCGA GATCCACATTGCCAGCAGAGCGGTCACGC-3′ for S146A; 5′- CCTTTGTTGT CTTCCTGGCAGAATTTGCCAGCAATAC-3′ and 5′- GTATTGCTGGCAAAT TCTGCCAGGAAGACAACAAAGG-3′ for T373A. E. coli BL21 (DE3) cells transformed with the expression vectors were grown in LB media to an attenuance of 0.5 at 600 nm and induced with 0.5 mM IPTG at 30 °C for 4 h. Cells were collected by centrifugation and ruptured by multiple passages through a pre-cooled French pressure cell. All the membrane protein purification experiments were conducted at 4 °C. Membranes were collected by ultracentrifugation (100,000 g for 2 h) and extracted with 1% (wt/vol) n-dodecyl-β-maltoside (DDM, Anatrace) in 20 mM HEPES-NaOH pH7.5, 100 mM NaCl, 20% (vol/vol) glycerol and 1 mM tris(2-carboxyethyl)phosphine (TCEP). The soluble fraction was loaded onto Ni-NTA resin in 20 mM Hepes-NaOH pH7.5, 100 mM NaCl, 25% glycerol, 0.02% DDM and 1 mM TCEP. Protein was eluted using the same buffer supplemented with 500 mM imidazole. The protein sample was promptly desalted and incubated with thrombin overnight. After thrombin cleavage the protein sample was desalted and concentrated to ~20 mg ml$^{-1}$ before it was further purified by using gel filtration chromatography (Superdex 200) in 20 mM Hepes-NaOH pH7.5, 100 mM NaCl, 100 mM sodium succinate or citrate, 10% glycerol, 0.14% (wt/vol) n-decyl-β-maltoside (DM, Anatrace) and 1 mM TCEP. For proteoliposome reconstitution, DDM was used throughout the protein purification and no succinate or citrate was added.

**Protein crystallization and crystal derivatization.** Prior to crystallization, VcINDY or MT5 was concentrated to ~10 mg ml$^{-1}$ and dialyzed extensively against 100 mM NaCl, 100 mM sodium succinate or citrate (pH~7), 20% glycerol, 0.14% DM and 1 mM TCEP at 4 °C. Crystallization experiments were performed using the hanging-drop vapor-diffusion method at 22 °C. The protein samples were mixed with equal volume of a crystallization solution containing 100 mM NaCl, 100 mM sodium succinate or citrate (pH~7), 36–40% (wt/vol) PEG1000, 0.14% DM, 20% glycerol and 1 mM TCEP. Protein crystals usually appeared within two weeks and continued to grow to full size in a month. For heavy atom derivatization, protein crystals were incubated with 20–50 mM heavy metal compounds for >5 h at 22 °C.

**Structure determination and refinement.** X-ray diffraction data were collected on the frozen crystals at the beam-lines 23-ID and 22-ID at Argonne National Laboratory. X-ray data were processed using the programme suite HKL2000 (ref. [49]) and further analysed using the CCP4 package[50] unless specified otherwise. All the structures were solved by combining molecular replacement and MIRAS phasing. The protein model (PDB 4F35) was placed into the unit cell by using the programme PHASER[51]. Heavy metal-binding sites were identified by difference Fourier analysis and MIRAS phases were calculated using the programme SHARP[52]. The resulting electron density maps were improved by solvent flattening, histogram matching, non-crystallographic symmetry averaging and phase extension. Model building was carried out using the programme O (ref. [53]). Structure refinement was conducted using the programme REFMAC with experimental phases as restraints[54].

**Proteoliposome reconstitution and transport assay.** DDM-purified VcINDY variants were reconstituted into liposomes at 0.1 µg µl$^{-1}$ using Escherichia coli polar lipids and 1-palmitoyl-2-oleoylphosphatidylcholine at a 3:1 (w/w) ratio[13,55]. The internal solution of liposomes contained 20 mM Tris-HEPES, pH 7.4, 30 mM K$_2$-succinate, 3 mM NaCl and 140 mM KCl. The transport assay was performed at 20 °C. Specifically, the reaction solution for the $K_M/V_{max}$ measurement in cases of VcINDY and MT5 consisted of 20 mM Tris-HEPES, pH 7.4, 0-2 mM K$_2$-succinate, 200 mM NaCl, 4 µM Valinomycin and 0.05-0.3 mM [$^{14}$C]-succinate (Moravek, CA). To analyse the Na$^+$-dependence of succinate transport by VcINDY, the reaction solution contained 20 mM Tris-HEPES, pH 7.4, 0–200 mM NaCl, 4 µM Valinomycin and 0.05 mM [$^{14}$C]-succinate. The reaction conditions were varied for S146A and T373A, which were specified in the figure captions. For all reactions, at 1-min time point, which lay well within the linear portion of the transport curve, a 60-µl sample was withdrawn and diluted in 0.5 ml of ice-cold quench buffer consisting of 100 mM Tris-HEPES, pH 7.4, 0-800 mM NaCl (same as that of the reaction buffer) and 1 mM EDTA. The quenched reaction was immediately filtrated through a nitrocellulose membrane (0.22 µm pore size, Millipore) and followed by three swift washes each with 1 ml of ice-cold quench buffer. Filters were dissolved in acetic acid and the radioactivity was measured by liquid scintillation. Transport assays were performed in duplicate and repeated more than three times.

**Data availability.** The coordinates of the models and structure factors have been deposited into the PDB data bank under the accession codes 5UL7, 5UL9, 5ULD and 5ULE. The PDB structure model 4F35 was used in this work. All the other data supporting the findings of this study are available within the article and its supplementary information files are available from the corresponding author upon reasonable request.

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

## Acknowledgements

We thank the beam-line staff at GM/CA-CAT (23-ID) and SER-CAT (23-ID) of Argonne National Laboratory for assistance during X-ray data collection. We also thank M. Radchenko, J. Mayor and E. Walters for their help during the early stages of this

project, and M. Glucksman for critical reading of the manuscript. This work was supported in part by the National Institutes of Health (R01-GM094195 to M.L.).

## Author contributions

M.L. conceived of the study. R.N. conducted site-directed mutagenesis, expressed the proteins and prepared the membranes. M.L. purified and crystallized the proteins. M.L. and J.S. collected and analysed the X-ray diffraction data. S.S. reconstituted the proteoliposomes and developed the transport assay. R.N. performed the transport kinetic studies. R.S.K. supervised the transport kinetic studies. M.L. wrote the manuscript.

## Additional information

**Competing interests:** The authors declare no competing financial interests.

