## [Peer Review File · Nature Communications]

Reviewer #1 (Remarks to the Author)

Nie R et al. present a very thorough and careful study of the vcINDY transporter of the divalent anion/Na⁺ symporter (DASS) family that transports dicarboxylates and is inhibited by citrate. Structures represent an inward substrate-bound state and two bound Na⁺ ions are identified. The study not only provides compelling evidence to support important corrections of previously published details at the citrate and Na⁺ binding sites (from structures that overall were otherwise correct, and using also those previously published data), but present also new input to understand citrate binding in the related human citrate transporter NaCT using an octuple mutant construct. Mechanistic models of Na⁺ symport and citrate inhibition/adaptation are presented and include also previously published studies. Not least the extensive functional characterisation by the Mindell-lab is very helpful, and largely consistent results are obtained on Na⁺ dependence, transport rates and the Hill coefficient.

Technically the study is carefully done and supports the presented manuscript well. The crystallographic studies are almost perfectionistic in their systematic use of multiple heavy-metal derivatives, but given the fact that corrections of previously published data are always a bit controversial and difficult to settle in full, and that the detailed studies are addressed at medium resolution, one can only approve of this level of care to make the study fully conclusive as presented.

Overall the study appears highly valuable and suitable for publication and the following remarks are only minor suggestions.

1. As mentioned above, the authors could comment if/why experimental phasing (which was used extensively for all structures) was in fact required in all cases or only of confirmatory value. It is an important message to the community on how membrane protein crystallography should be conducted both conclusively and efficiently and model-based phases should not be disregarded if fully sufficient.
2. On page 10 the authors may perhaps refer to LeuBAT as another well-documented case of a multi-mutational approach to "humanise" a bacterial transporter (<https://www.ncbi.nlm.nih.gov/pubmed/24121440>)
3. On page 14 the comparison to other anion transporters highlights that DASS do not use Lys/Arg residues unlike other transporters, but it is based on passive description only. However from a mechanistic point of view, it is not clear if this has to be so for DASS and can be explained, or if it merely happens to be so with various compensations of charges/amino acid side chains (as described). Can other Na⁺ sites be proposed? Mulligan et al. & Mindell suggest at least 3 and also here the functional data indicate 3 Na⁺ or more.
4. An Arg occupies the Na⁺/substrate site in Glu/Asp transporters in the apo state (<https://www.ncbi.nlm.nih.gov/pubmed/24013209>; <https://www.ncbi.nlm.nih.gov/pubmed/24842876>) and it would be very interesting to include such considerations of DASS and the Na⁺ sites (cf. 3 above). How would DASS/vcINDY deal with a substrate free state and return to the outward facing state? Can anything be learnt from a comparison to the related AbgT antiporters/exchangers that solve this by an antiport mechanism?

Reviewer #2 (Remarks to the Author)

The submitted manuscript, Nie et al.'s "Structure and function of the divalent anion/Na⁺ symporter from *Vibrio cholera* and a humanized variant thereof," offers a fascinating view into the structural basis of substrate, inhibitor, and sodium recognition within a bacterial (and humanized variant) of the DASS family transporter, VcINDY. In contrast to previous work, the authors were able to assign placement of both Na⁺ ions (supported by biochemical analysis), substrate and inhibitor density, rebuild an erroneous portion of a previous citrate-bound model, and present a humanized model offering possible insights into the eukaryotic function of DASS transporters. The manuscript is thorough, and the author's core findings seem supported by their data.

DASS transporters shuffle Krebs cycle intermediates across cell membranes, and as such are of particular importance in metabolism through the direct regulation of ATP production, impacting

adipose deposition and even lifespan in model organisms. Eukaryotic DASS transporters are thought to couple transport of carboxylates such as citrate or succinate with three or more Na⁺ ions via a mechanism of symport. Bacterial homologs are known to couple transport of succinate with at least two Na⁺ ions, however citrate functions in these systems as a competitive inhibitor of substrate uptake. Previous structural work on the bacterial DASS family transporter VcINDY established a general model for understanding these critical transporters, and curiously was found to be stabilized in the presence of citrate. Notably, only one Na⁺ binding site was identified in this model.

Building on this previous framework, the authors were able to generate a co-crystallized structure of VcINDY with succinate. The improved resolution and electron density of this model revealed a 10 amino acid stretch of improperly built protein, which critically was found to harbor the second Na⁺ binding site. The authors' spatial location of this site is not only supported by their experimental density, but also by mutation/functional abatement marking this a significant find. Furthermore, these mutational studies in conjunction with the succinate-bound model allowed the authors to elucidate a means of Na⁺ binding site/substrate recognition/stabilization mechanics.

The authors also present a citrate-bound VcINDY co-crystal structure, which not only revealed further errors in the initial placement of the molecule in the previous structure, but also in doing so revealed that the binding site (precluding the terminal carboxylate) overlaps significantly with succinate. This provides a clear example of citrate's mode of co-inhibition, by precluding succinate binding at the recognition site.

While the manuscript is interesting and presents many important new findings, a number of amendments should be addressed before publication:

Major revisions

1. The introduction and discussion leave the reader confused as to how the current work fits into, and builds on, what is already structurally and functionally known from previous work. Substantial clarification in this regard to help place the current work into the context of the literature is required.
2. A section devoted to the thorough structural comparison of the current structure vs the previous structure should be made. Please report rmsds overall and applicable crystallographic information.

Minor revisions

1. The Na⁺ sites should both be further validated by submitting the structures to the CheckMyMetal (CMM): Metal Binding Site Validation Server. Please discuss the result.
2. Please remove objectionable statements such as 'strategies for extending healthspan in humans' and 'lend fresh hope for meeting acute medical needs'. These statements are a stretch when reporting a bacterial transporter structure.
3. Please overlay the rebuilt parts of the structure and colour by either C alpha rmsd movement or relative b-factor differences. Please include this as a supplemental figure.
4. Please discuss why it was necessary to collect multiple heavy metal soak datasets for all structures. It may be helpful to discuss the benefits of such a strategy in terms of avoiding molecular replacement map bias, as the reported experimental electron density maps are indeed of excellent quality.
5. Please remove the shadows from Figure 2.
6. Please keep the colours for succinate and citrate consistent in figures 5 6 and 8 as well as in the supplemental figures.

Reviewers' comments:

Reviewer #1 (Remarks to the Author):

Nie R et al. present a very thorough and careful study of the vcINDY transporter of the divalent anion/Na⁺ symporter (DASS) family that transports dicarboxylates and is inhibited by citrate. Structures represent an inward substrate-bound state and two bound Na⁺ ions are identified. The study not only provides compelling evidence to support important corrections of previously published details at the citrate and Na⁺ binding sites (from structures that overall were otherwise correct, and using also those previously published data), but present also new input to understand citrate binding in the related human citrate transporter NaCT using an octuple mutant construct. Mechanistic models of Na⁺ symport and citrate inhibition/adaptation are presented and include also previously published studies. Not least the extensive functional characterisation by the Mindell-lab is very helpful, and largely consistent results are obtained on Na⁺ dependence, transport rates and the Hill coefficient.

Technically the study is carefully done and supports the presented manuscript well. The crystallographic studies are almost perfectionistic in their systematic use of multiple heavy-metal derivatives, but given the fact that corrections of previously published data are always a bit controversial and difficult to settle in full, and that the detailed studies are addressed at medium resolution, one can only approve of this level of care to make the study fully conclusive as presented.

Overall the study appears highly valuable and suitable for publication and the following remarks are only minor suggestions.

Response: We sincerely thank Reviewer#1 for taking the time to review our manuscript and for the overall positive comment. We have addressed the points raised by Reviewer#1 as follows.

1. As mentioned above, the authors could comment if/why experimental phasing (which was used extensively for all structures) was in fact required in all cases or only of confirmatory value. It is an important message to the community on how membrane protein crystallography should be conducted both conclusively and efficiently and model-based phases should not be disregarded if fully sufficient.

Response: We have revised the manuscript to discuss the importance of experimental phasing in our work (page 4, first paragraph).

2. On page 10 the authors may perhaps refer to LeuBAT as another well-documented case of a multi-

mutational approach to "humanise" a bacterial transporter
(<https://www.ncbi.nlm.nih.gov/pubmed/24121440>)

Response: We have cited this work in the revised manuscript (page 12, first paragraph).

3. One page 14 the comparison to other anion transporters highlights that DASS do not use Lys/Arg residues unlike other transporters, but it is based on passive description only. However from a mechanistic point of view, it is not clear if this has to be so for DASS and can be explained, or if it merely happens to be so with various compensations of charges/amino acid side chains (as described). Can other Na⁺ sites be proposed? Mulligan et al. & Mindell suggest at least 3 and also here the functional data indicate 3 Na⁺ or more.

Response: We have expanded our discussion section to clarify the point regarding the lack of Lys/Arg in VcINDY (page 16, second paragraph).

In terms of the additional Na⁺-binding site, thus far we have no crystallographic or biochemical data to guide us to locate such site. We think the identification of the additional Na⁺-binding site may require the structure determination of VcINDY or its homologue captured in a different conformational state, as well as the structure-based mutational study, which are beyond the scope of this work.

4. An Arg occupies the Na⁺/substrate site in Glu/Asp transporters in the apo state (<https://www.ncbi.nlm.nih.gov/pubmed/24013209>; <https://www.ncbi.nlm.nih.gov/pubmed/24842876>) and it would be very interesting to include such considerations of DASS and the Na⁺ sites (cf. 3 above). How would DASS/vcINDY deal with a substrate free state and return to the outward facing state? Can anything be learnt from a comparison to the related AbgT antiporters/exchangers that solve this by an antiport mechanism?

Response: Excellent suggestions, we have included discussion about the apo state of VcINDY in our manuscript (page 16, second paragraph).

To our knowledge, the substrate-binding site in AbgT has not been established (refs. 39 and 40). Therefore, it is not feasible for us to discuss the transport mechanism of AbgT with that of VcINDY in detail.

Reviewer #2 (Remarks to the Author):

The submitted manuscript, Nie et al.'s "Structure and function of the divalent anion/Na⁺ symporter from *Vibrio cholera* and a humanized variant thereof," offers a fascinating view into the structural basis of substrate, inhibitor, and sodium recognition within a bacterial (and humanized variant) of the DASS family transporter, VcINDY. In contrast to previous work, the authors were able to assign placement of both Na⁺ ions (supported by biochemical analysis), substrate and inhibitor density, rebuild an erroneous

portion of a previous citrate-bound model, and present a humanized model offering possible insights into the eukaryotic function of DASS transporters. The manuscript is thorough, and the author's core findings seem supported by their data.

DASS transporters shuffle Krebs cycle intermediates across cell membranes, and as such are of particular importance in metabolism through the direct regulation of ATP production, impacting adipose deposition and even lifespan in model organisms. Eukaryotic DASS transporters are thought to couple transport of carboxylates such as citrate or succinate with three or more Na⁺ ions via a mechanism of symport. Bacterial homologs are known to couple transport of succinate with at least two Na⁺ ions, however citrate functions in these systems as a competitive inhibitor of substrate uptake. Previous structural work on the bacterial DASS family transporter VcINDY established a general model for understanding these critical transporters, and curiously was found to be stabilized in the presence of citrate. Notably, only one Na⁺ binding site was identified in this model.

Building on this previous framework, the authors were able to generate a co-crystallized structure of VcINDY with succinate. The improved resolution and electron density of this model revealed a 10 amino acid stretch of improperly built protein, which critically was found to harbor the second Na⁺ binding site. The authors' spatial location of this site is not only supported by their experimental density, but also by mutation/functional abatement marking this a significant find. Furthermore, these mutational studies in conjunction with the succinate-bound model allowed the authors to elucidate a means of Na⁺ binding site/substrate recognition/stabilization mechanics.

The authors also present a citrate-bound VcINDY co-crystal structure, which not only revealed further errors in the initial placement of the molecule in the previous structure, but also in doing so revealed that the binding site (precluding the terminal carboxylate) overlaps significantly with succinate. This provides a clear example of citrate's mode of co-inhibition, by precluding succinate binding at the recognition site.

While the manuscript is interesting and presents many important new findings, a number of amendments should be addressed before publication:

Response: We deeply thank Reviewer#2 for commenting on our manuscript and for making excellent suggestions. We have revised our manuscript accordingly, which is detailed below.

Major revisions

1. The introduction and discussion leave the reader confused as to how the current work fits into, and builds on, what is already structurally and functionally known from previous work. Substantial clarification in this regard to help place the current work into the context of the literature is required.

Response: We have revised our manuscript to clarify this point (page 3, first and second paragraphs).

2. A section devoted to the thorough structural comparison of the current structure vs the previous structure should be made. Please report rmsds overall and applicable crystallographic information.

Response: We have included such discussion in our manuscript (page 10, second paragraph).

Minor revisions

1. The Na⁺ sites should both be further validated by submitting the structures to the CheckMyMetal (CMM): Metal Binding Site Validation Server. Please discuss the result.

Response: We have included this analysis in our manuscript (page 5, first paragraph).

2. Please remove objectionable statements such as ‘strategies for extending healthspan in humans’ and ‘lend fresh hope for meeting acute medical needs’. These statements are a stretch when reporting a bacterial transporter structure.

Response: We have removed these statements and revised the text accordingly (abstract; page 14, first paragraph).

3. Please overlay the rebuilt parts of the structure and colour by either C alpha rmsd movement or relative b-factor differences. Please include this as a supplemental figure.

Response: We have included this new figure in the manuscript (current Supplementary Figure 4).

4. Please discuss why it was necessary to collect multiple heavy metal soak datasets for all structures. It may be helpful to discuss the benefits of such a strategy in terms of avoiding molecular replacement map bias, as the reported experimental electron density maps are indeed of excellent quality.

Response: We have included this discussion in the manuscript (page 4, first paragraph).

5. Please remove the shadows from Figure 2.

Response: We have remade the figure as suggested (current Figure 2).

6. Please keep the colours for succinate and citrate consistent in figures 5 6 and 8 as well as in the supplemental figures.

Response: We have remade the figures as suggested (current Figures 6, 8 and Supplementary Figure 8).

Reviewer #1 (Remarks to the Author)

The authors have addressed the reviewer comments well, although I still have some minor comments

1. The very extensive use of heavy-metal derivatives is of course excellent when available, and it has for sure been important for a first structure determination and model correction in this presented series of structures, but but it will have required a very significant use of time where the added value may have been in very small increments, and it is important that e.g. students and unexperienced crystallographers reading the paper are not left with the impression that only a puristic procedure of experimental phasing for ALL structures will be reliable strategy. It would be helpful to add a further comment (top of page 4 where this previous point has been addressed already) on which structures for which the MIRAS phases were particular helpful as compared to molecular replacement. Something like ".....Although not essential for solving the phase problem, the MIRAS phases allowed us to model substantially more amino acids than what had been achieved previously¹⁹ (445 vs. 398 out of 462 residues) and to locate the bound succinate with confidence. Moreover, the density-modified MIRAS maps revealed that parts of the published VcINDY structure¹⁹, which are directly related to the binding of Na⁺ and citrate, had been incorrectly determined (see below). Using the updated model and difference maps other structures were well determined by as also supported by unbiased MIRAS maps"

2. Further candidate Na⁺ sites probably emerge from model analysis (e.g. pinpointing Asp/Glu residues placed near to e.g. Ser/Thr/Tyr/Asn/Gln in TM locations) and can be further qualified by e.g. sequence conservation and mutational studies. Are there really no candidates for further Na⁺ sites?

REVIEWERS' COMMENTS:

Reviewer #1 (Remarks to the Author):

The authors have addressed the reviewer comments well, although I still have some minor comments

1. The very extensive use of heavy-metal derivatives is of course excellent when available, and it has for sure been important for a first structure determination and model correction in this presented series of structures, but but it will have required a very significant use of time where the added value may have been in very small increments, and it is important that e.g. students and unexperienced crystallographers reading the paper are not left with the impression that only a puristic procedure of experimental phasing for ALL structures will be reliable strategy. It would be helpful to add a further comment (top of page 4 where this previous point has been addressed already) on which structures for which the MIRAS phases were particular helpful as compared to molecular replacement. Something like ".....Although not essential for solving the phase problem, the MIRAS phases allowed us to model substantially more amino acids than what had been achieved previously¹⁹ (445 vs. 398 out of 462 residues) and to locate the bound succinate with confidence. Moreover, the density-modified MIRAS maps revealed that parts of the published VcINDY structure¹⁹, which are directly related to the binding of Na⁺ and citrate, had been incorrectly determined (see below). Using the updated model and difference maps other structures were well determined by as also supported by unbiased MIRAS maps"

Response: We have revised the text as suggested by the reviewer.

2. Further candidate Na⁺ sites probably emerge from model analysis (e.g. pinpointing Asp/Glu residues placed near to e.g. Ser/Thr/Tyr/Asn/Gln in TM locations) and can be further qualified by e.g. sequence conservation and mutational studies. Are there really no candidates for further Na⁺ sites?

Response: We appreciate the reviewer's suggestions. However, it is not feasible for us to propose one or two potential Na⁺-binding sites for a number of reasons. Firstly, neither Na1 nor Na2 contains a Asp or Glu, thus it is conceivable that the unobserved Na⁺-binding site(s) may also lack such charged residue(s). Therefore, it is not clear if we can identify the missing site(s) by searching for Asp/Glu within the transmembrane domain. Secondly, Na1 and Na2 both involve backbone carbonyl oxygens. Thus it is likely that such backbone atoms coordinate Na⁺ in the unobserved site(s). In other words, the side-chains of some of the Na⁺-binding amino acids in VcINDY may not be conserved since only backbone atoms are utilized in cation coordination. Based on these considerations, without available biochemical data, it is extremely challenging, if not impossible, for us to identify potential Na⁺-binding site(s) in VcINDY. As such, we have chosen to be circumspect and refrain from suggesting new Na⁺-binding site(s) on the basis of our structures. We hope that this treatment would be acceptable to the reviewer.

We also asked for Reviewer #2 to comment on the revised version of your manuscript. This reviewer privately stated that the issues raised were addressed.